# A Unified First-Order Framework for Activation Steering and Data Influence

## Abstract

*Activation steering* adds a low-dimensional vector to an intermediate layer of a neural network to elicit or suppress behaviors, whereas *influence functions* trace the effect of infinitesimally re-weighting training examples on model outputs. We prove that, to first order, these techniques are *equivalent*: any steering vector can be represented as an influence weighting over training data and vice versa. This duality yields: (i) a constructive algorithm for mapping undesired behaviors back to causal training examples; (ii) an optimal-control perspective on steering that reveals its regularization properties; and (iii) generalization bounds for low-rank steering interventions. Our analysis adds theoretical clarity to two popular but previously disconnected strands of interpretability research.

## 1 Introduction

Large-scale neural networks—exemplified by transformer language models, diffusion–based image generators, and vision transformers—have become indispensable across science, industry, and culture. Their success, however, stands in tension with two practical desiderata. First, behavioral steering: practitioners often wish to suppress toxicity, reveal internal reasoning, or insert new factual knowledge without the prohibitive cost of retraining billions of parameters. Second, causal attribution: when a model exhibits bias or hallucination, we would like to trace that behavior to the specific training examples that gave rise to it. Current toolkits address these goals along two largely independent lines.

**Activation steering.** This family of methods keeps the learned weights fixed and instead injects a low-dimensional vector into an intermediate layer during inference (Subramani et al., 2022). Activation-space steering has been used to detoxify harmful or biased language outputs (Turner et al., 2023; Wang & Shu, 2024), compress or elicit chain-of-thought reasoning (Azizi et al., 2025), flip or erase specific factual memories via knowledge neurons (Dai et al., 2022), and robustly edit whole fact distributions with SAKE's optimal-transport activation edits (Scialanga et al., 2025). See also (Zou et al., 2023). Because it modifies only activations, steering is fast, does not disturb the original checkpoint, and can be toggled on or off per query.

**Training-data influence.** Influence-function techniques tackle attribution from the opposite end. By differentiating the empirical loss twice, they estimate how infinitesimally up-weighting a single training example would have altered today's prediction (Koh & Liang, 2017). The resulting influence scores underpin modern workflows for dataset debugging, bias auditing, and dataset distillation. See also (Pruthi et al., 2020; Barshan et al., 2020; Toneva et al., 2019; Feldman & Zhang, 2020).

Although both lines of work pursue model *controllability*, their operational spaces are orthogonal: activation steering assumes frozen weights, whereas influence analysis assumes fixed activations and perturbs the weights that produced them. Practitioners therefore face an unsatisfying dichotomy: experiment blindly with steering and, if it fails, resort to expensive parameter interventions—without guidance on *when* steering can succeed or *how* to connect a successful steering vector back to its causal data.

We show that these two perspectives are, to first order, *projections of the same underlying sensitivity tensor*. Concretely, we construct an **Influence-Aligned Steering** (IAS) vector that, for any infinitesimal influence re-weighting, induces an identical logit shift—and we prove the converse mapping

from steering to influence. This equivalence is not merely conceptual: it yields explicit diagnostic and optimization tools that scale to billion-parameter models.

**Scope and empirical justification.** We focus on the *small-edit* regime used in practice. First-order analysis yields closed-form constructions (IAS), principal-angle diagnostics ($\gamma$), and predictable compute. Empirically, predicted and realized logit shifts are nearly collinear for small edits (cosine $\approx 0.98$; Fig. 1). For compact weight-space adaptation, see (Hu et al., 2022; Aghajanyan et al., 2021).

1. Steer–influence equivalence. We establish a closed-form duality that maps every steering perturbation to a signed influence measure over the training set, and vice versa.

2. Alignment-based feasibility. A single scalar $\gamma(x)$—the cosine of the smallest principal angle between two Jacobian subspaces—fully characterizes when perfect equivalence is possible. If $\gamma(x)$ is small, we prove a no-free-lunch lower bound showing that no activation-space edit can replicate the effect of data re-weighting.

3. Spectral Optimality. Given a norm budget, the steering direction that maximizes first-order logit change is the leading eigenvector of a Fisher–influence matrix; this spectral recipe replaces hand-crafted vectors.

4. Practical workflow. All quantities (Section 5) reduce to Jacobian–vector products and pseudoinverses, requiring only two backward passes per input. Practitioners can therefore (i) prototype with steering, (ii) identify the responsible training examples, and (iii) decide—with $\gamma$—whether weight-level editing is necessary.

By unifying steering and influence under one first-order lens, IAS offers a single, efficient workflow for controllability and data provenance.

## 2 BACKGROUND AND NOTATION

**A running toy example.** See Appendix C for a compact linear-network illustration of IAS.

**Model and layer of interest.** Let $f_{\boldsymbol{\theta}} : \mathcal{X} \to \mathbb{R}^m$ be a network with parameters $\boldsymbol{\theta} \in \mathbb{R}^P$ and logits $f_{\boldsymbol{\theta}}(x)$. Fix a layer of width $d$ with pre-activations $\mathbf{h}(x) \in \mathbb{R}^d$. We use the Jacobians

$$\mathbf{J}_{h \to y}(x) := \frac{\partial f_{\boldsymbol{\theta}}(x)}{\partial \mathbf{h}(x)} \in \mathbb{R}^{m \times d}, \quad \mathbf{J}_{\theta \to y}(x) := \frac{\partial f_{\boldsymbol{\theta}}(x)}{\partial \boldsymbol{\theta}} \in \mathbb{R}^{m \times P}, \quad \mathbf{J}_{\theta \to h}(x) := \frac{\partial \mathbf{h}(x)}{\partial \boldsymbol{\theta}} \in \mathbb{R}^{d \times P}.$$

**Assumptions.** (i) *Feasibility:* when stated, $\mathrm{Im}(\mathbf{J}_{\theta \to y}) \subseteq \mathrm{Im}(\mathbf{J}_{h \to y})$ so IAS exists and is unique; (ii) *Local smoothness:* a $\kappa$-Lipschitz neighborhood for Jacobians (Cor. 2); (iii) *Affine independence:* for $\ell_1$-minimality of $\rho_{\mathbf{s}}$ in Cor. 1.

**Notation.** $\mathcal{S}_h(x) := \mathrm{Im}(\mathbf{J}_{h \to y}(x))$ and $\mathcal{S}_\theta(x) := \mathrm{Im}(\mathbf{J}_{\theta \to y}(x))$ are subspaces of logit space; $\gamma(x) := \cos \angle_{\min}(\mathcal{S}_\theta, \mathcal{S}_h) \in [0, 1]$ is their smallest principal-angle cosine; $\mathbf{F}_h := \mathbf{J}_{h \to y} \mathbf{J}_{h \to y}^\top$ is the activation-Fisher.

**Influence functions.** Let $\ell(z, \boldsymbol{\theta})$ be the per-example loss and $\mathbf{H}_{\boldsymbol{\theta}} := \nabla_{\boldsymbol{\theta}}^2 \frac{1}{|\mathcal{Z}|} \sum_{z \in \mathcal{Z}} \ell(z, \boldsymbol{\theta})$ the *empirical Hessian (or its damped Gauss–Newton surrogate)*, assumed positive-(semi)definite on a relevant subspace. Up-weighting a training point $z$ by $\epsilon \ll 1$ induces $\Delta\boldsymbol{\theta}_z = -\epsilon \mathbf{H}_{\boldsymbol{\theta}}^{-1} \nabla_{\boldsymbol{\theta}} \ell(z, \boldsymbol{\theta})$, and the first-order logit shift on test input $x$ is

$$\Delta y^{\mathrm{IF}}(x) = \mathbf{J}_{\theta \to y}(x) \, \Delta\boldsymbol{\theta}_z. \tag{1}$$

Define the per-example first-order logit influence as $\mathcal{I}(z \to x) := \mathbf{J}_{\theta \to y}(x) \Delta\boldsymbol{\theta}_z$ (cf. Eq. equation 1). We use a damped inverse $(\mathbf{H}_{\boldsymbol{\theta}} + \lambda I)^{-1}$ for stability (Appendix D.1). In all experiments, $\lambda > 0$ is treated as a Tikhonov regularizer; $\mathbf{H}$ may be replaced by a Gauss–Newton approximation without changing the first-order theory.

**Computational primitives (cost model).** All results rely on: (i) two Jacobian–vector or vector–Jacobian products per input, (ii) a rank-$d$ pseudoinverse of $\mathbf{J}_{h\to y}$ (never larger than the layer width), and (iii) a small SVD to estimate principal angles for $\gamma$.

**Activation steering.** Adding $\alpha\mathbf{s} \in \mathbb{R}^d$ at the chosen layer yields the logit shift

$$\Delta y^{\text{SV}}(x) = \mathbf{J}_{h\to y}(x)\,(\alpha\mathbf{s}). \tag{2}$$

Equations equation 1–equation 2 share a linear form; the remainder of the paper characterizes when one can stand in for the other and how to construct the corresponding perturbation efficiently.

## 3 A DUAL VIEW: PARAMETER–ACTIVATION SENSITIVITIES

**Why add one more lens?** We have already seen that two linear maps govern first-order behavior: the parameter–logit Jacobian $\mathbf{J}_{\theta\to y}$ and the activation–logit Jacobian $\mathbf{J}_{h\to y}$. Theorems 5.1–6.2 will quantify their interaction, but first we show that the maps form a *primal–dual* pair in the convex-analysis sense.

**Two complementary projections.** The primal view is an *orthogonal projection* of the desired logit displacement $\mathbf{J}_{\theta\to y}\Delta\theta$ onto $\mathcal{S}_h(x)$, then a lift back to activation space with minimum energy. The dual view projects in the *Fisher norm* induced by activations; the dual multiplier $\boldsymbol{\lambda}^\star$ is the Fisher-metric certificate of effort required to cover components outside $\mathcal{S}_h(x)$.

**Rule of thumb.** If $\|\boldsymbol{\lambda}^\star\|$ is small, steering is cheap and faithful; if large, a weight-space update is likely necessary. Computing $\boldsymbol{\lambda}^\star$ is as cheap as IAS itself (two JVPs), so the check can precede any search for directions.

We start from the *inverse* problem: given a desired parameter-space displacement $\Delta\theta$ (e.g., an influence update), find the shortest activation change that reproduces its logit effect.

### 3.1 THE PRIMAL PROGRAM: LEAST-EFFORT STEERING

$$\min_{\Delta\mathbf{h}\in\mathbb{R}^d} \frac{1}{2}\|\Delta\mathbf{h}\|_2^2 \quad \text{s.t.} \quad \mathbf{J}_{h\to y}\,\Delta\mathbf{h} = \mathbf{J}_{\theta\to y}\,\Delta\theta. \tag{P}$$

**Feasibility.** If $\text{Im}(\mathbf{J}_{\theta\to y}) \subseteq \text{Im}(\mathbf{J}_{h\to y})$, the constraint is feasible and the Euclidean minimum-norm solution exists and is unique.

### 3.2 THE DUAL PROGRAM

Introduce $\boldsymbol{\lambda} \in \mathbb{R}^m$. Minimizing the Lagrangian over $\Delta\mathbf{h}$ yields $\Delta\mathbf{h}^\star = \mathbf{J}_{h\to y}^\top\boldsymbol{\lambda}^\star$ with

$$\boldsymbol{\lambda}^\star = -\big(\mathbf{J}_{h\to y}\mathbf{J}_{h\to y}^\top\big)^\dagger\mathbf{J}_{\theta\to y}\,\Delta\theta, \qquad \Delta\mathbf{h}^\star = \mathbf{J}_{h\to y}^\dagger\,\mathbf{J}_{\theta\to y}\,\Delta\theta. \tag{2}$$

Thus the *Influence-Aligned Steering (IAS)* vector is the projection of the target logit movement onto the activation-reachable subspace, lifted back with the Moore–Penrose pseudoinverse.

**Geometry and diagnosis.** $\mathbf{F}_h := \mathbf{J}_{h\to y}\mathbf{J}_{h\to y}^\top$ is the Fisher information of the logits w.r.t. activations; $\boldsymbol{\lambda}^\star$ is the Fisher-metric certificate of effort. A large $\|\boldsymbol{\lambda}^\star\|$ signals that most of the desired displacement lies outside the activation subspace and that steering will require large energy (or fail), anticipating the alignment bounds below.

## 4 STEERING–INFLUENCE DUALITY AT THE DATA LEVEL

The primal–dual view explains the existence of an optimal steering vector for a *given* parameter perturbation. Related scalable data-attribution methods include (Pruthi et al., 2020; Barshan et al., 2020). We now climb one level up and ask for a direct correspondence between steering interventions and *training-data* re-weightings.

**Lemma 4.1** (Chain-rule factorization). *For any scalar metric $m_{\boldsymbol{\theta}}(x)$ and any layer $\ell$,*

$$\nabla_{\boldsymbol{\theta}}\, m_{\boldsymbol{\theta}}(x) \;=\; J_{\boldsymbol{\theta}\to h^{(\ell)}}^{\top}\, \nabla_{h^{(\ell)}} m_{\boldsymbol{\theta}}(x).$$

*Sketch.* Differentiate $m_{\boldsymbol{\theta}}$ along the composite map $\boldsymbol{\theta}\to\mathbf{h}^{(\ell)}\to m_{\boldsymbol{\theta}}$. $\qquad\square$

**Theorem 4.2** (Steering–Influence Equivalence). *Let $\mathbf{s}\in\mathbb{R}^d$ be added with magnitude $\alpha\ll 1$ at layer $\ell$. There exists a signed measure $\rho_{\mathbf{s}}$ over the training set such that*

$$f_{\boldsymbol{\theta}}^{\mathbf{s},\alpha}(x) - f_{\boldsymbol{\theta}}(x) = \sum_{z\in\mathcal{Z}} \rho_{\mathbf{s}}(z)\,\mathcal{I}(z\to x) \;+\; O(\alpha^2), \quad \|\rho_{\mathbf{s}}\|_1 = |\alpha|. \tag{4}$$

*Conversely, any signed weighting $\mathbf{w}\in\mathbb{R}^{|\mathcal{Z}|}$ with $\|\mathbf{w}\|_1 = \epsilon$ admits a steering vector $\mathbf{s}_{\mathbf{w}}$ with $\|\mathbf{s}_{\mathbf{w}}\| = O(\epsilon)$ that realizes the same first-order output shift.*

**Residual when spans do not match.** If $\mathrm{Im}(\mathbf{J}_{h\to y})$ does not contain $\mathrm{Im}(\mathbf{J}_{\theta\to y})$, perfect matching is impossible. Writing $P_h$ for the orthogonal projection onto $\mathcal{S}_h(x)$, the irreducible residual obeys

$$\left\|(I - P_h)\,\mathbf{J}_{\theta\to y}\Delta\theta\right\|_2 \;\le\; \sqrt{1-\gamma(x)^2}\,\left\|\mathbf{J}_{\theta\to y}\Delta\theta\right\|_2, \tag{3}$$

the logit-space version of Theorem 5.1. In practice, we use equation 3 as a pre-check: small $\gamma(x) \Rightarrow$ skip steering.

The result holds exactly if the set $\{\mathcal{I}(z\to x)\}_{z\in\mathcal{Z}}$ spans $\mathrm{Im}(\mathbf{J}_{h\to y})$; otherwise Eq. equation 4 holds up to a residual whose norm is bounded by $\left(1 - \gamma(x)^2\right)^{1/2}\|\alpha\mathbf{s}\|$.

**Intuition.** Equation 4 says that a steer vector $\alpha\mathbf{s}$ acts like redistributing $|\alpha|$ units of mass across training examples, weighted by how well their gradients correlate with $\mathbf{s}$. The minimal-$\ell_1$ measure that achieves this correlation is precisely $\rho_{\mathbf{s}}$.

**Implication.** Given an empirical steering direction, the associated measure $\rho_{\mathbf{s}}$ points straight to the *most causal* training documents. In practice, one inspects the top-weighted examples to debug bias or privacy leaks.

### 4.1 FROM STEERING TO DATA: A CAUSAL COROLLARY

**Corollary 1** (Minimal data re-weighting induced by steering). *Assume that the influence vectors $\{\mathcal{I}(z\to x)\}_{z\in\mathcal{Z}}$ are affinely independent; otherwise the $\ell_1$-minimal solution need not be unique. Let $(\mathbf{s},\alpha)$ be an activation-space intervention at layer $\ell$ with $\|\mathbf{s}\| = 1$ and $|\alpha|\ll 1$. Among all signed measures $\nu$ on the training set that reproduce the first-order logit shift,*

$$\Delta y^{\mathrm{SV}}(x) \;=\; \sum_{z\in\mathcal{Z}} \nu(z)\,\mathcal{I}(z\to x),$$

*the measure $\rho_{\mathbf{s}}$ constructed in Eq. 4 is $\ell_1$-minimal, i.e. $\|\rho_{\mathbf{s}}\|_1 \;=\; \min_{\nu}\big\{\|\nu\|_1 \;:\; \nu$ satisfies the equation$\big\} = |\alpha|$.*

*Idea of the proof.* Equation 4 already realizes the shift with $\|\rho_{\mathbf{s}}\|_1 = |\alpha|$. If another measure $\nu$ achieved the same shift with smaller $\ell_1$ norm, one could scale $\rho_{\mathbf{s}}$ down and still match the shift, contradicting the definition of $\alpha$ as the steering magnitude. $\qquad\square$

**Practical payoff.** Given an empirical steering vector, $\rho_{\mathbf{s}}$ pinpoints the *fewest* training examples to relabel/remove/examine to reproduce the behavioral change (see Section 7).

### 4.2 A GEOMETRIC PICTURE OF ALIGNMENT

Let $\mathcal{S}_{\theta}(x) := \mathrm{Im}(\mathbf{J}_{\theta\to y}(x))$ and $\mathcal{S}_h(x) := \mathrm{Im}(\mathbf{J}_{h\to y}(x))$ be subspaces in logit space. The primal program P orthogonally projects $\mathbf{J}_{\theta\to y}\Delta\boldsymbol{\theta}$ onto $\mathcal{S}_h(x)$ and lifts to the minimum-norm activation; the dual equation 2 performs the projection in the Fisher norm $\mathbf{F}_h := \mathbf{J}_{h\to y}\mathbf{J}_{h\to y}^{\top}$. Small principal angles imply close projections and modest $\|\boldsymbol{\lambda}^{\star}\|$; near-orthogonality yields the no-free-lunch regime.

**Practical diagnostic.** The norm of $\boldsymbol{\lambda}^\star$ quantifies unreachable components: small $\|\boldsymbol{\lambda}^\star\|$ implies faithful, low-energy steering; large values suggest weight-space editing. Computing $\boldsymbol{\lambda}^\star$ costs two JVP/VJPs (same as IAS), enabling a quick steer-vs-retrain decision.

**Choosing the layer $\ell$ in practice.** Across LMs we find (Fig. 2) that $\gamma$ typically increases toward later blocks. A simple heuristic is therefore: probe $\gamma$ at a few candidate layers on a small prompt batch and pick the smallest layer index with $\gamma \geq 0.7$.

This balances headroom (later layers) with locality (earlier layers).

## 5 Main Theoretical Guarantees

### 5.1 When does steering perfectly match influence?

**Theorem 5.1** (Alignment Bound). *For any infinitesimal parameter perturbation $\Delta\boldsymbol{\theta}$, the relative logit error of the minimum-norm IAS vector $\Delta\mathbf{h}^\star$ satisfies*

$$\frac{\|\mathbf{J}_{\theta\to y}\Delta\boldsymbol{\theta} - \mathbf{J}_{h\to y}\Delta\mathbf{h}^\star\|_2}{\|\mathbf{J}_{\theta\to y}\Delta\boldsymbol{\theta}\|_2} \leq \sqrt{1 - \gamma^2(x)},$$

*where $\gamma(x)$ is the cosine of the smallest principal angle between the column spaces of $\mathbf{J}_{h\to y}(x)$ and $\mathbf{J}_{\theta\to y}(x)$ (Björck & Golub, 1973).*

*Intuition and use.* Overlap (large $\gamma$) enables exact matching; misalignment limits fidelity at rate $\sqrt{1 - \gamma^2}$. Computing $\gamma$ (two small SVDs) quickly certifies feasibility.

### 5.2 The unique steering vector if alignment holds

**Theorem 5.2** (Minimum-Norm IAS). *If $\mathrm{Im}(\mathbf{J}_{\theta\to y}) \subseteq \mathrm{Im}(\mathbf{J}_{h\to y})$, the unique steering vector that solves problem equation P is*

$$\Delta\mathbf{h}^\star = \mathbf{J}_{h\to y}^\dagger \mathbf{J}_{\theta\to y} \Delta\boldsymbol{\theta}.$$

*Note.* This is the orthogonal projection/lift solution; two JVP/VJPs and a rank-$\leq d$ pseudoinverse suffice in practice.

**Corollary 2** (Second-order radius). *If the map $\boldsymbol{\theta} \mapsto \big(\mathbf{J}_{\theta\to y}, \mathbf{J}_{\theta\to h}\big)$ is $\kappa$-Lipschitz in a neighborhood of $\boldsymbol{\theta}$, then the Taylor remainder obeys $\|f_{\boldsymbol{\theta}+\Delta\boldsymbol{\theta}} - f_{\boldsymbol{\theta}} - \mathbf{J}_{\theta\to y}\Delta\boldsymbol{\theta}\|_2 \leq \kappa\|\Delta\boldsymbol{\theta}\|_2^2$, and the matching IAS perturbation incurs the same $O(\alpha^2)$ error.*

### 5.3 Steering maximally under an $\ell_2$ budget

**Theorem 5.3** (Spectral Optimality). *Fix a norm budget $\|\mathbf{s}\| \leq B$. Let*

$$\Sigma := \frac{1}{|\mathcal{Z}|}\sum_{z\in\mathcal{Z}} \mathbf{J}_{\theta\to h}^\top \mathbf{H}_{\boldsymbol{\theta}}^{-1}\nabla_{\boldsymbol{\theta}}\ell(z,\boldsymbol{\theta})\,\nabla_{\boldsymbol{\theta}}\ell(z,\boldsymbol{\theta})^\top \mathbf{H}_{\boldsymbol{\theta}}^{-1}\mathbf{J}_{\theta\to h}.$$

*The steering vector that maximizes the expected first-order logit change is the top eigenvector $\mathbf{s}_{\max}$ of $\Sigma$, and the achievable change equals $B\sqrt{\lambda_{\max}(\Sigma)}\,\|\nabla_{\mathbf{h}}f_{\boldsymbol{\theta}}(x)\|$.*

**Estimating the spectral direction (practical recipe).** Power iteration with Hutchinson-style mini-batches suffices:

1. Initialize $v_0 \sim \mathcal{N}(0, I_d)$.

2. For $t = 0, 1, \ldots$: draw a mini-batch $\mathcal{B}$; compute $g_z := \mathbf{J}_{\theta\to h}^\top(\mathbf{H} + \lambda I)^{-1}\nabla_{\theta}\ell(z,\theta)$ for $z \in \mathcal{B}$; set $v_{t+1} \propto \sum_{z\in\mathcal{B}} g_z(g_z^\top v_t)$.

3. Stop when $\|v_{t+1} - v_t\|/\|v_t\| < \varepsilon$; return $v_t$.

*Note.* $\Sigma$ averages influence correlations; its top eigenvector gives a principled steering direction estimated via one power-iteration over mini-batches.

**Lemma 5.4** (Layer-wise composability). *Let $\gamma_1, \gamma_2$ be the alignment cosines for two consecutive layers. Applying IAS at layer 1 and layer 2 yields a combined alignment cosine at least*

$$\gamma_{12} \; \geq \; \gamma_1\,\gamma_2 \; = \; \sqrt{1 - \left(1 - \gamma_1^2\right)}\,\sqrt{1 - \left(1 - \gamma_2^2\right)}.$$

*Consequently, mis-alignment compounds multiplicatively.*

## 6 GENERALIZATION UNDER LOW-RANK STEERING

**Theorem 6.1** (Rademacher-complexity blow-up under rank-$k$ steering). *Let $f_{\boldsymbol{\theta}}$ be the base model and $\tilde{f} = f_{\boldsymbol{\theta}} + \alpha UV^{\top}$ the model obtained by adding a rank-$k$ IAS correction at layer $\ell$, with $\|U\|_2 = \|V\|_2 = 1$. For any loss $\ell$ that is L-Lipschitz in its first argument, the empirical Rademacher complexity satisfies*

$$\mathfrak{R}_n(\ell \circ \tilde{f}) \; \leq \; \mathfrak{R}_n(\ell \circ f_{\boldsymbol{\theta}}) \; + \; \alpha L\,\sqrt{\frac{2k}{dn}},$$

*where $d$ is the width of layer $\ell$ and $n$ the sample size.*

*Sketch.* Combine Thm. 2 of Pinto et al. (2024) with the fact that IAS changes only a rank-$k$ sub-matrix of the layer weight. The additional Rademacher term is bounded by $\alpha L\sqrt{2k/dn}$. □

**From complexity to risk.** Let $\widehat{\mathcal{L}}$ be the empirical risk and $\mathcal{L}$ the population risk. With probability $1 - \delta$,

$$\mathcal{L}(\tilde{f}) - \mathcal{L}(f_{\boldsymbol{\theta}}) \; \leq \; 2\,\mathfrak{R}_n(\ell \circ \tilde{f}) + c\sqrt{\frac{\log(1/\delta)}{n}} \; \lesssim \; 2\,\mathfrak{R}_n(\ell \circ f_{\boldsymbol{\theta}}) + 2\alpha L\sqrt{\frac{2k}{dn}} + c\sqrt{\frac{\log(1/\delta)}{n}}, \quad (4)$$

for a universal constant $c$. Thus, for fixed budget $\alpha$ and modest rank $k \ll d$, the excess risk term due to IAS vanishes as $d$ and $n$ grow.

**Practical guidance.** (i) Prefer low ranks $k$ and smaller $\alpha$ unless $\gamma$ is close to 1. (ii) When $\gamma < 0.5$, skip steering and switch to weight-space editing; the bound equation 3 predicts poor fidelity. (iii) Treat damping $\lambda$ as a regularizer that trades a small bias for numerical stability in $\mathbf{H}^{-1}$ (Appendix D.1).

### 6.1 WHEN STEERING IS *provably* INSUFFICIENT

**Theorem 6.2** (No-Free-Lunch). *Let $\gamma(x)$ denote the cosine of the smallest principal angle between $\mathrm{Im}(J_{\theta \to y}(x))$ and $\mathrm{Im}(J_{h \to y}(x))$. If $\gamma(x) \leq \rho < 1$, then for every activation perturbation $\Delta \mathbf{h}$ and the corresponding (best-possible) parameter perturbation $\Delta\boldsymbol{\theta}$ we have*

$$\frac{\left\|J_{h \to y}(x)\,\Delta \mathbf{h}\right\|_2}{\left\|J_{\theta \to y}(x)\,\Delta\boldsymbol{\theta}\right\|_2} \; \leq \; \gamma(x) \; \leq \; \rho.$$

*Intuition.* Poor alignment means the desired logit displacement lives largely outside the steering subspace; even an infinite-norm activation change cannot push further than factor $\rho$.

*Consequence for practice.* If the quick diagnostic yields a small $\gamma(x)$, engineers can skip steering entirely and proceed straight to parameter-space editing.

IAS is the exact minimum-energy activation edit matching a target first-order logit displacement; its fidelity is controlled by $\gamma(x)$. The spectral recipe provides a principled way to choose a strong direction under a budget, and low-rank IAS has a benign impact on generalization. When $\gamma$ is small, the geometry itself forbids steering to fully replace influence.

## 7 EXPERIMENTS

**Setup.** Unless stated otherwise we use GPT-2 Medium and steer at layer $\ell=8$. Steering vectors are built from 50 toxic vs. 50 neutral Jigsaw prompts; evaluation uses 500 TOXIGEN prompts. Toxicity is scored with DETOXIFY; perplexity is measured on a benign WikiText subset.

## 7.1 LANGUAGE-MODEL DETOXIFICATION VIA STEERING

We compare Contrastive Activation Addition (CAA) with our Influence-Aligned Steering (IAS), using identical $\ell_2$ magnitude and layer. Table 1 reports mean toxicity (lower is better) and benign-PPL.

|  | Baseline | CAA | IAS |
|---|---|---|---|
| Toxicity (mean) ↓ | 0.0195 | **0.0150** | 0.0164 |
| Perplexity ↓ | 14333 | **13291** | 13701 |

Table 1: Detoxification on 500 TOXIGEN prompts with benign-PPL on WikiText (GPT-2 Medium, $\ell=8$).

## 7.2 FIRST-ORDER EQUIVALENCE: IAS MATCHES INFLUENCE AT FIRST ORDER

Our theory predicts that the *first-order* logit shift from an influence update is matched by the minimum-norm IAS vector. Over $n=5000$ prompt–token pairs at $\ell=8$, predicted vs. actual shifts are nearly collinear (cosine 0.978, slope 1.50), consistent with the expected linear regime.

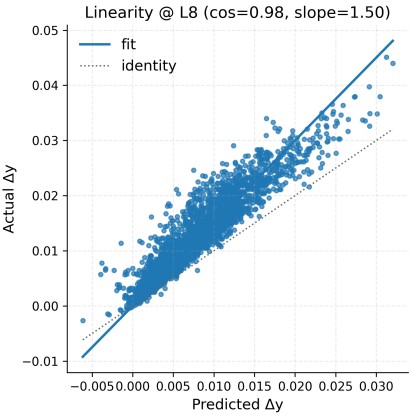

Figure 1: **IAS $\approx$ influence (first order).** Predicted (first-order) vs. actual logit shifts for $n=5000$ pairs at $\ell=8$; cosine 0.978, slope 1.50.

## 7.3 ALIGNMENT VS. LAYER DEPTH

The feasibility diagnostic $\gamma(x)$ increases with depth on GPT-2 Medium, with the median rising from 0.64 at layer 0 to 0.94 by layer 11 (Figure 2). This supports Theorem 5.1: late layers provide the best subspace overlap for steering to match influence.

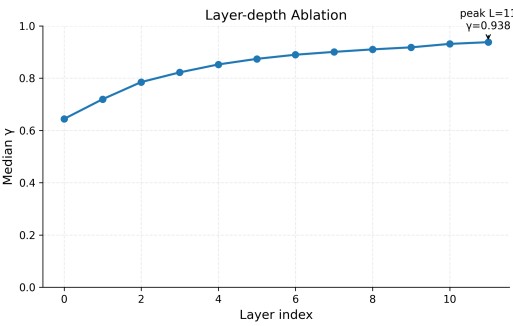

Figure 2: **Layer-depth ablation of alignment.** Median $\gamma$ across prompts monotonically increases from 0.64 (L0) to 0.94 (L11).

### 7.4 SPECTRAL OPTIMALITY OF STEERING DIRECTIONS (IMAGENET)

We test the vision analog of Theorem 5.3 on ResNet-50 by estimating the spectral direction that maximizes the horse logit (class 339). Figure 3 compares the spectral shift against random directions: the spectral radius lies far in the tail of the null distribution ($p=0.00498$, $z=3.55$).

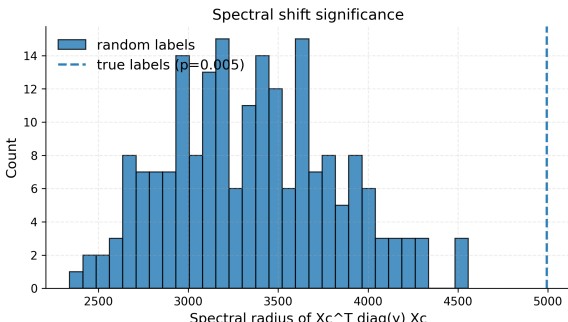

Figure 3: **Spectral shift significance** (ResNet-50, horse class). Dashed line: spectral direction; histogram: random directions.

## 8 RELATED WORK

Activation steering originated in sentiment control for language models (Turner et al., 2023) and has since grown into a family of latent-direction methods. Influence functions were ported from classical statistics to deep nets by Koh & Liang (2017). Our work is the first to give a closed-form map between the two ideas and to quantify when one subsumes the other. Concurrent work on parameter-space editing (ROME (Meng et al., 2022), MEMIT (Meng et al., 2023)) tackles a complementary regime: finite, non-infinitesimal changes to factual knowledge.

## 9 CONCLUSION

We have shown that steering vectors and influence functions—previously separate tools—live on the same geometric plane. Influence-Aligned Steering provides the mathematical bridge, complete with error guarantees, constructive formulas, and impossibility results. Beyond its theoretical appeal, IAS promises an integrated workflow for debugging, auditing, and aligning large neural models: steer first, trace provenance, edit weights only when the geometry demands it.

IAS is a first-order theory; very large steering magnitudes or influence perturbations beyond the quadratic regime may violate the linear approximation. Extending the analysis to second order—where Hessian–Jacobian interactions appear—is left for future work. Moreover, computing exact pseudoinverses is tractable for single layers but challenging for deep stacks; exploring Krylov or randomized SVD methods is an open engineering problem.

**AI assistance disclosure.** We used large language models to polish grammar and improve the clarity of some sentences.

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
