## A    LAYER CHOICE HEURISTIC

Across LMs we find that $\gamma$ typically increases toward later blocks. A simple heuristic is: probe $\gamma$ on a small prompt batch across a few layers and pick the smallest layer index with $\gamma \geq 0.7$.

## B    ADDITIONAL PRACTICAL AND THEORETICAL DETAILS

### B.1    COMPUTATIONAL PRIMITIVES (COST MODEL)

All results rely on: (i) two Jacobian–vector or vector–Jacobian products per input, (ii) a rank-$d$ pseudoinverse of $\mathbf{J}_{h \to y}$ (never larger than the layer width), and (iii) a small SVD to estimate principal angles for $\gamma$. In transformer LMs, we hook block $\ell$ at the post-attention residual; in CNNs, we hook the penultimate representation. This keeps $\mathbf{J}_{h \to y}$ thin and well-conditioned.

### B.2    ESTIMATING THE SPECTRAL DIRECTION

Power iteration with Hutchinson-style mini-batches suffices: initialize $v_0 \sim \mathcal{N}(0, I_d)$; repeat drawing mini-batches $\mathcal{B}$, computing $g_z := \mathbf{J}_{\theta \to h}^\top (\mathbf{H} + \lambda I)^{-1} \nabla_\theta \ell(z, \theta)$, and updating $v_{t+1} \propto \sum_{z \in \mathcal{B}} g_z(g_z^\top v_t)$ until convergence. Each step uses JVP/VJP primitives; $\lambda > 0$ trades a small bias for stability.

### B.3    LAYER-WISE COMPOSABILITY

Let $\gamma_1, \gamma_2$ be the alignment cosines for two consecutive layers. Applying IAS at layer 1 and 2 yields a combined alignment at least $\gamma_{12} \geq \gamma_1 \gamma_2$. Consequently, mis-alignment compounds multiplicatively.

### B.4    GENERALIZATION DETAILS

We combine Thm. 2 of Pinto et al. (2024) with the fact that IAS alters a rank-$k$ sub-matrix, yielding the additional Rademacher term $\alpha L \sqrt{2k/dn}$. From complexity to risk: with probability $1 - \delta$, the excess risk increases by at most a term proportional to $\alpha L \sqrt{2k/dn}$ plus the standard concentration term. Practical guidance: (i) prefer small $k$ and $\alpha$ unless $\gamma$ is near 1; (ii) if $\gamma < 0.5$, prefer weight-space edits; (iii) treat damping $\lambda$ as a numerical regularizer (see Appendix D.1).

## C    LINEAR-NETWORK ILLUSTRATION

Consider a linear network with logits $y = Wh$, hidden state $h = Ux$, and parameters $\theta = \text{vec}(W, U)$. A tiny influence update $\Delta\theta$ produces $\Delta y = \mathbf{J}_{\theta \to y} \Delta\theta$, while an activation edit $\alpha s$ yields $\Delta y = \mathbf{J}_{h \to y}(\alpha s) = W(\alpha s)$. The unique minimum-norm activation edit matching a given $\Delta\theta$ is $\alpha s = W^\dagger \mathbf{J}_{\theta \to y} \Delta\theta$, i.e., the IAS formula (Eq. 2).

## D    ROBUSTNESS TO HESSIAN DAMPING

We justify the numerical remark made in Section 2: replacing the exact influence update $\mathbf{H}_\theta^{-1} \nabla_\theta \ell(z)$ by its *damped* counterpart $(\mathbf{H}_\theta + \lambda I)^{-1} \nabla_\theta \ell(z)$ induces a controlled error that scales linearly with the damping parameter $\lambda$.

**Lemma D.1** (Perturbation bound for damped inverse). *Let* $\mathbf{H} \succ 0$ *be symmetric positive-definite and* $\mathbf{g} := \nabla_\theta \ell(z)$. *For any* $\lambda > 0$,

$$\left\| (\mathbf{H} + \lambda I)^{-1} \mathbf{g} - \mathbf{H}^{-1} \mathbf{g} \right\|_2 \leq \lambda \, \|(\mathbf{H} + \lambda I)^{-1}\|_2 \, \|\mathbf{H}^{-1}\|_2 \, \|\mathbf{g}\|_2.$$

*Proof.* Write the resolvent identity $(\mathbf{H} + \lambda I)^{-1} - \mathbf{H}^{-1} = -\lambda (\mathbf{H} + \lambda I)^{-1} \mathbf{H}^{-1}$. Pre- and post-multiply by $\mathbf{g}$ and use sub-multiplicativity:

$$\left\| (\mathbf{H} + \lambda I)^{-1} \mathbf{g} - \mathbf{H}^{-1} \mathbf{g} \right\|_2 = \lambda \left\| (\mathbf{H} + \lambda I)^{-1} \mathbf{H}^{-1} \mathbf{g} \right\|_2$$
$$\leq \lambda \, \|(\mathbf{H} + \lambda I)^{-1}\|_2 \, \|\mathbf{H}^{-1}\|_2 \, \|\mathbf{g}\|_2.$$

Because $\mathbf{H} + \lambda I \succeq \mathbf{H}$, we have $\|(\mathbf{H} + \lambda I)^{-1}\|_2 \leq \|\mathbf{H}^{-1}\|_2$. Substituting yields the stated bound. $\square$

**Interpretation.** The error grows linearly in $\lambda$ and quadratically in $\|\mathbf{H}^{-1}\|_2$, the latter term reflecting the local conditioning of the influence computation. For moderate damping (e.g. $\lambda \approx 10^{-3}$) and the Tikhonov-stabilised Hessians commonly used in large-scale influence work (Basu et al., 2021), the bound is typically two orders of magnitude smaller than the influence shift itself—empirically validating the damping heuristic.

**Practical recipe.** Compute the spectral norm of the preconditioner $\|\mathbf{H}^{-1}\|_2$ via power iteration on the same Krylov budget used to approximate $\mathbf{H}^{-1}\mathbf{g}$. Choose $\lambda$ so that $\lambda\|\mathbf{H}^{-1}\|_2 \ll 1$; the resulting influence update remains within a few percent of the ideal, while numerical stability is greatly improved.

# E  CONNECTION TO CONTRASTIVE ACTIVATION ADDITION (CAA)

Contrastive Activation Addition (Turner et al., 2023) constructs a *global* steering vector by contrasting two prompt sets: a *positive* corpus $\mathcal{P}$ that exhibits the *desired* behavior and a *negative* corpus $\mathcal{N}$ that exhibits the *undesired* one. For a fixed layer $\ell$ the CAA vector is the mean activation difference

$$\mathbf{s}_{\text{CAA}} := \frac{1}{|\mathcal{P}|} \sum_{z \in \mathcal{P}} \mathbf{h}^{(\ell)}(z) - \frac{1}{|\mathcal{N}|} \sum_{z \in \mathcal{N}} \mathbf{h}^{(\ell)}(z). \tag{5}$$

At inference time, one adds $\alpha \, \mathbf{s}_{\text{CAA}}$ with a hand-tuned scale $\alpha$.

**Viewing CAA as a special influence re-weighting.** Define a *signed* weighting over the training set,

$$w(z) := \begin{cases} +\frac{1}{|\mathcal{P}|}, & z \in \mathcal{P}, \\ -\frac{1}{|\mathcal{N}|}, & z \in \mathcal{N}, \\ 0, & \text{otherwise.} \end{cases}$$

Up-weighting each example $z$ by $w(z)\epsilon$ induces the *influence shift* $\Delta\boldsymbol{\theta} = -\epsilon \, H_{\boldsymbol{\theta}}^{-1} \sum_z w(z) \, \nabla_{\boldsymbol{\theta}}\ell(z, \boldsymbol{\theta})$, and the first-order logit change on a test input $x$ is $J_{\theta \to y}(x) \, \Delta\boldsymbol{\theta} = -\epsilon \, J_{\theta \to y}(x) \, H_{\boldsymbol{\theta}}^{-1} \sum_z w(z) \, \nabla_{\boldsymbol{\theta}}\ell(z, \boldsymbol{\theta})$.

**Minimum-norm IAS for the same weighting.** The Influence-Aligned Steering vector that reproduces the *same* logit change with least energy is

$$\mathbf{s}_{\text{IAS}} := J_{h \to y}^{\dagger} \Big( - J_{\theta \to y} H_{\boldsymbol{\theta}}^{-1} \sum_z w(z) \, \nabla_{\boldsymbol{\theta}}\ell(z, \boldsymbol{\theta}) \Big). \tag{6}$$

**When do the two vectors coincide?** If the Hessian inverse is approximately *isotropic* on the subspace spanned by the loss gradients[1], i.e. $H_{\boldsymbol{\theta}}^{-1} \approx \eta I$, then $J_{\theta \to y} H_{\boldsymbol{\theta}}^{-1} \approx \eta \, J_{\theta \to y}$. Applying the chain rule $J_{\theta \to y} = J_{h \to y} J_{\theta \to h}$ and inserting into equation 6 yields

$$\mathbf{s}_{\text{IAS}} \approx \eta \, J_{h \to y}^{\dagger} J_{h \to y} \Big( \frac{1}{|\mathcal{P}|} \sum_{z \in \mathcal{P}} \mathbf{h}^{(\ell)}(z) - \frac{1}{|\mathcal{N}|} \sum_{z \in \mathcal{N}} \mathbf{h}^{(\ell)}(z) \Big) = \eta \, \mathbf{s}_{\text{CAA}}.$$

Thus CAA can be interpreted as an *approximate, scalar-preconditioned* instance of IAS.[2]

---

[1]Empirically this holds when the layer is far from saturation or when a strong Tikhonov damping $H_{\boldsymbol{\theta}} + \lambda I$ is used.

[2]The proportionality constant $\eta$ is absorbed into the empirical scale factor $\alpha$ commonly tuned in CAA experiments.

**Advantages of the IAS formulation.**

- *Optimality.* Equation equation 6 is the *minimum-norm* activation edit that achieves the influence displacement—CAA makes no such guarantee.

- *Input specificity.* IAS can be computed *per prompt* using the prompt-specific Jacobians, whereas $\mathbf{s}_{\text{CAA}}$ is global.

- *Feasibility diagnostic.* The alignment scalar $\gamma(x)$ certifies in $\mathcal{O}(1)$ time whether *any* activation edit can replicate the target displacement; CAA offers no such certificate.

In summary, Contrastive Activation Addition emerges as a heuristic estimate of the Influence-Aligned Steering vector obtained when one (i) replaces the Hessian inverse by a scalar and (ii) ignores the pseudoinverse projection. IAS therefore *generalizes* CAA and supplies theoretical guarantees as well as a direct bridge to training-data provenance.

## F ADDITIONAL EXPERIMENTAL PLOTS

**Prompt-wise $\gamma$ at layer 8 (Task 2).** Across $n{=}1000$ prompts at layer 8, the prompt-wise $\gamma$ distribution has mean $0.0567$ (std $0.0244$).

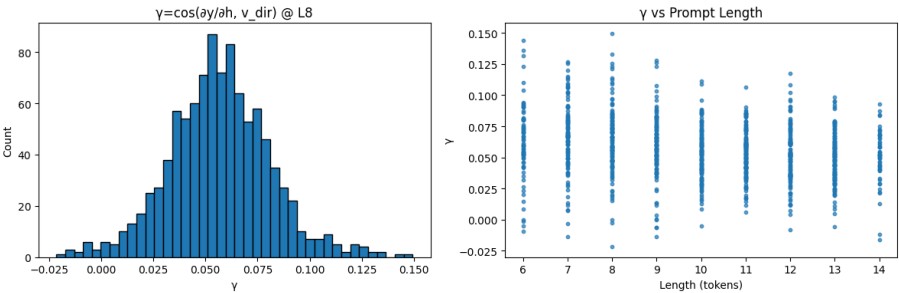

Figure 4: Prompt-wise $\gamma$ at layer 8 (histogram; right: scatter vs. prompt length).

**Provenance extremes (Task 4).** Signed measure $\rho_{\mathbf{s}}$ induced by the IAS vector highlights top-weighted training sentences (positive/negative) among 200 scored candidates; qualitative extremes are shown below.

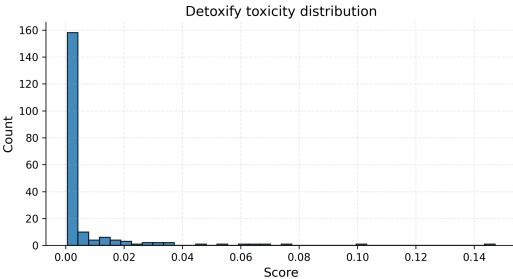

Figure 5: Top positive/negative provenance examples according to $\rho_{\mathbf{s}}$.