# OpenReview forum: "A Unified First‑Order Framework for Activation Steering and Data Influence"
_ICLR.cc/2026/Conference — ICLR 2026 Conference Withdrawn Submission_

### Official Review · Reviewer_a5Ni · 2025-10-29

**Soundness:** 2
**Presentation:** 3
**Contribution:** 2
**Rating:** 4
**Confidence:** 3

**Summary:**

The authors present a mathematical connection between influence functions (data influences on model outputs) and activation steering vectors (vectors added to activations to encourage behavioral characteristics). The authors use this mathematical connection as a bridge between two seemingly different fields of interpretability research. In addition to presenting a mathematical equivalence between the two and where the equivalence holds, they provide practical results showing how to evaluate when the bridge exists and empirical results on gpt2 medium to verify their theory.

**Strengths:**

- bridging two important areas of mech interp under one theory is a profound contribution even if the bridge does not always hold
- they provide an analysis of cases where the equivalence holds and where it breaks down
- they defend their theory with empirical simulations
- they provide practical insight (how to practically determine when steering and influence functions have an equivalence and how to use their results to make steering vectors)

**Weaknesses:**

Overall, my main concerns with this paper is that the claims in the abstract are not supported by their findings, and the paper is difficult to follow mathematically (although that may be a shortcoming of my own).

Additional specific suggestions/concerns:
- would be good to be explicit about the approximate nature of equations 1 and 2 when they are introduced. I believe those equations only hold if $f_\theta(x)$ is linear or if $\alpha s$ and $\Delta \theta$ are very small, neither of which are true for your settings. The equation that you are approximating is an integration along a path from $h$ to $\alpha s$ (and the equivalent for $\Delta t$)
- no justification for gpt2 medium and layer 8
- the claims in the abstract and claim 1 in the intro need to be toned down. As written, it sounds like a perfect equivalence always holds, which is not true, as you claim in claim 2 in the intro.
- the spectral optimality finding is not useful in cases where you do not know the precise logits that you want to influence but rather a high-level concept/function (e.g. toxicity)
- in general, I believe many of the analyses rest on the assumption that the goal of steering is to change the logits by a specific amount with well defined inputs, which is not always true. Another goal of steering is to affect higher level behaviors under novel inputs. I could be misunderstanding some of the ideas presented in the paper, but this needs to be addressed.

**Questions:**

- What is contrastive activation addition (CAA) in Section 7.1?
- if I'm understanding correctly, the math is all performed for a single data input? Does everything hold for a batch of samples? I do not believe it does, because $(W1+W2)x \neq Wx+s$ for multiple vectors x. So, the equivalence does not hold for multiple, individual inputs?

---

### Official Review · Reviewer_FR5K · 2025-10-31

**Soundness:** 1
**Presentation:** 1
**Contribution:** 2
**Rating:** 2
**Confidence:** 4

**Summary:**

This paper introduces Influence-Aligned Steering (IAS), a method that connects activation steering with influence functions. The authors argue that any activation steering direction can be interpreted as an influence-weighted combination of training examples, and conversely, that influence functions can generate effective steering vectors. This theoretical bridge is used to unify activation editing, weight editing, and data attribution under a single conceptual framework. The paper further proposes practical guidelines on choosing between activation-level and weight-level interventions based on this connection.

**Strengths:**

- Presents an interesting idea that connects activation editing to weight editing, and further to training data attribution.
- Proposes a unifying first-order perspective that links steering, Jacobians, feasibility, and spectral structure. If validated, this framework could help standardize diagnostics for "steerability."

**Weaknesses:**

- The main text and appendix do not provide proofs for core claims, making it impossible to verify correctness. At a minimum, the paper should include complete statements with assumptions and provide proof sketches in the main paper and full proofs in the appendix. Otherwise, results should be clearly labeled as conjectural.
- Several symbols are undefined or ambiguous. For example, in line 86, $\mathbf{h}(x)$ is referred to as “pre-activations,” but it is unclear what this means—does $\mathbf{h}$ represent the concatenated pre-activations across all layers or a specific layer? The meaning of $y$ is also undefined. A concise notation table would improve clarity.
- There are several issues with the equations. For example: $H_\theta$ in line 101 doesn’t appear to account for $\epsilon$; $\Delta \mathbf{h}^*$ in line 147 is missing a minus sign; and equation numbering is inconsistent (e.g., 1, 2, P, 2, 4, 3, 4).
-  The inclusion $Im(J_{\theta \to y}) \subseteq Im(J_{h \to y})$ is a strong and currently unjustified claim. Please provide the conditions under which this inclusion holds, or offer counterexamples where it fails.
- The experimental section only evaluates GPT-2 Medium, which is insufficient to support claims about modern large language models. At least one contemporary 7B-scale model, such as from the LLaMA or Gemma families, should be included.
- In Table 1, the proposed method performs worse than CAA, a simpler technique. This discrepancy should be addressed with analysis or discussion.
- The paper only explores a limited form of activation steering: vector addition at a single layer. It’s unclear how the proposed method generalizes to more complex settings, such as steering at multiple layers or using more advanced steering methods like directional ablation [1,2] or rotational transformations [3,4].
- A more thorough review of the activation steering and data attribution literature is recommended.

[1] Refusal in Language Models Is Mediated by a Single Direction

[2] Representation Engineering: A Top-Down Approach to AI Transparency

[3] Angular Steering: Behavior Control via Rotation in Activation Space

[4] Householder Pseudo-Rotation: A Novel Approach to Activation Editing in LLMs with Direction-Magnitude Perspective

**Questions:**

See Weaknesses

---

### Official Review · Reviewer_h1dP · 2025-10-31

**Soundness:** 3
**Presentation:** 2
**Contribution:** 2
**Rating:** 6
**Confidence:** 1

**Summary:**

This work proposes a first order mapping between activation steering and influence functions, demonstrating the equivalence between two prominently used methods in model controllability. They propose an Influence-Aligned Steering vector to reproduce the same first order logit shift as any influence reweighting, and additionally prove the converse mapping holds. The empirical results across multiple datasets demonstrate consistency with the proposed theory.

**Strengths:**

1. The work provides a clean mathematical bridge between action space and data level influence, offering a closed form mapping that is novel accelerates the field of model controllability.
2. The intuition, implication and practicality paragraphs in section 4 were very beneficial in understanding the practical meaning of the work.
3. The empirical results covered multiple data modalities demonstrating cross-domain applicability.

**Weaknesses:**

1. The equivalence hinges on the assumption of feasibility, local smoothness and affine independence, yet there is no discussion regarding the feasibility of these assumptions.
2. Empirical results consider only one language model and vision model, so it unclear how well their theory adapts beyond model types.
3. As pseudoinverses and SVDs can be unstable, confidence metrics for their results taken over multiple seeds and parameters is necessary but missing.

**Questions:**

1. How sensitive is IAS to the choices of pseudoinverse regularization?

---

### Official Review · Reviewer_RXbW · 2025-11-01

**Soundness:** 3
**Presentation:** 3
**Contribution:** 3
**Rating:** 6
**Confidence:** 1

**Summary:**

This paper proposes a framework connecting two major strands of interpretability research: activation steering (adding low-dimensional vectors to intermediate activations to modify behavior) and influence functions (tracing output sensitivity to infinitesimal changes in training-data weights). The authors show that, to first order, these are mathematically equivalent. Empirically, they demonstrate these relationships on GPT-2 Medium (for language-model detoxification) and ResNet-50 (for spectral optimality), showing strong correspondence between predicted and realized logit shifts.

**Strengths:**

1. **Practical implications**. The unified view leads to a clear, actionable workflow for practitioners (use steering first, diagnose feasibility, and only then resort to weight-level editing). Practical payoffs and intuitions are made clear throughout.
2. **Clarity and intuition**. The prose is concise and continuously ties theoretical constructs back to practical goals.
3. **Empirical grounding**. The paper validates its claims on real models (GPT-2, ResNet-50).
4. **Potentially impactful**. Could influence both interpretability and alignment communities by offering a shared geometric language.
5. **Conceptual unification**. Provides a theoretical bridge between two previously separate interpretability techniques (steering and influence), deepening our understanding of both.

**Weaknesses:**

1. **Limited empirical scope**. Experiments are minimal and rely on older models (GPT-2 Medium, ResNet-50) with small datasets; stronger validation on current architectures would bolster credibility.
2. **Thin empirical discussion**. Section 7 largely reports numbers and figures without much narrative interpretation.
3. **First-order limitation**. While explicitly acknowledged, the framework’s reliance on the linear regime may limit applicability.

**Questions:**

1. How stable is the first-order equivalence empirically as the steering magnitude grows? Is there a quantified threshold beyond which the linear assumption breaks down?
2. Given the practical workflow (steer, trace, edit), can the authors share or describe a real-world debugging example?
3. Did the authors consider models beyond GPT-2 and ResNet-50?

---

### Note · Authors · 2025-11-22

I have read and agree with the venue's withdrawal policy on behalf of myself and my co-authors.